# AUTOMATIC CALIBRATION AND ERROR CORRECTION FOR GENERATIVE LARGE LANGUAGE MODELS VIA PARETO OPTIMAL SELF-SUPERVISION

## ABSTRACT

Generative Large Language Models (LLMs) have demonstrated remarkable capabilities for a wide range of applications, but reducing ungrounded or erroneous responses remains a major growth area. Unlike traditional task-specific models, which can provide calibrated outputs through training, there is a lack of effective methods to estimate the confidence level of generative LLMs' responses. Such estimations are crucial for indicating potential errors and facilitating human-in-the-loop verification. An important source of calibration comes from expert-stipulated programmatic supervision, often available at low cost, but with limitations such as noise and coverage issues. In this paper, we introduce a Pareto optimal self-supervision framework that leverages available programmatic supervision to systematically calibrate LLM responses. This framework produces a risk score for every LLM response without additional manual effort. This is accomplished by learning a harmonizer model that aligns with LLM output and other weak supervision sources. The model assigns higher risk scores to more uncertain LLM responses, thus facilitating error correction. Experiments on standard relation extraction and classification tasks in biomedical and general domains demonstrate that the proposed risk score is highly correlated with the actual LLM error rate. By using a dynamic prompting strategy based on the risk score, we observed significant accuracy improvement for off-the-shelf LLMs, boosting GPT-3.5 results past state-of-the-art (SOTA) weak supervision model and GPT-4 results past SOTA supervised results on challenging evaluation datasets.

## 1  INTRODUCTION

Generative Large Language Models (LLMs) have evolved to become impressively powerful in recent developments (Zhao et al., 2023), with Generative Pretrained Transformer (GPT) models showing increasingly effective capabilities. The evolution from GPT-3 (Brown et al., 2020) to GPT-4 (OpenAI, 2023), along with the emergence of other LLMs such as PaLM (Chowdhery et al., 2022) and LLaMA (Touvron et al., 2023), has marked a significant leap in natural language understanding and problem-solving abilities. The generative nature of these models has led to their widespread adoption in numerous application fields. However, as indicated in Ji et al. (2023), challenges such as hallucination or erroneous responses persist, especially in domains requiring high accuracy and reliability, such as biomedical and healthcare.

Unfortunately, systematic tools for efficiently identifying hallucinations or estimating the confidence level of outputs are lacking. Since the outputs are free text, the intrinsic confidence score from the generative LLMs is often unavailable or poorly calibrated with respect to the desired target, particularly after applying reinforcement learning with human feedback (RLHF) (Ouyang et al., 2022), as noted by OpenAI (2023). Researchers have recently resorted to heuristic approaches (Dziri et al., 2021; Sun et al., 2023; Chen et al., 2023; Dhuliawala et al., 2023; Elaraby et al., 2023; Lei et al., 2023; Cao et al., 2023; Luo et al., 2023) as a compromise. A common method involves querying the LLM in various ways to estimate the answer's correctness (e.g., Manakul et al. (2023)). However, these approaches are computationally expensive, biased by the LLM itself, and not quantitative.

To address these issues, we propose a novel approach to calibrate LLM outputs and automatically identify error responses. As an early attempt to tackle the LLM error problem, we restrict ourselves to problems where the expected output can be categorized, such as classification in the simplest setting. The intuitions behind our method are:

1. Distilling LLMs into smaller networks leads to calibration via implicit label smoothing.
2. Incorporating independent noisy signals is guaranteed to enhance LLM performance.

A theoretical analysis of these intuitions is provided in Section 3.2.

Our proposed self-supervision framework is illustrated in Fig. 1. For a given input instance, the LLM outputs its answer, which is then compared with multiple heuristic labels from noisy weak supervision signals such as knowledge bases (KB) and regular expressions (RX). As the LLM and weak sources may not always align, our goal is to train a harmonizer network $h(x)$ that provides a probabilistic estimate of the answer's correctness. This process, importantly, does not involve any human reviewing or labeling.

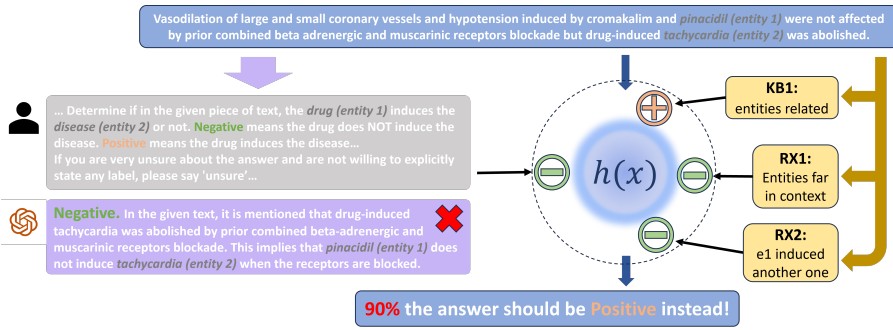

Figure 1: Self-supervision framework to calibrate LLM output and automatically detect error.

Previous works in programmatic weak supervision, which often combine multiple supervision sources into a single label per instance (Zhang et al., 2022; Wang & Poon, 2018), have shown success but also exhibit significant limitations when applied to identifying LLM errors, primarily due to the weighting dilemma. If the weight assigned to the LLM is too low, the aggregated result is noisy; if too high, the output is dominated by the LLM, making error detection difficult. Recent work by Rühling Cachay et al. (2021) addresses this with instance-dependent weighting, but the EM type of optimization faces significant learning variance.

In this paper, we approach the problem through multi-objective optimization, leveraging extensive research in Pareto optimization (Pareto, 1964). We introduce a flexible framework combining information from both the LLM response and weak supervision sources using Pareto optimal learning. The harmonizer network $h(x)$ is optimized on LLM and weak sources simultaneously in a Pareto optimal manner, effectively overcoming the weighting dilemma. Our key contributions are:

1. We are the first to propose adopting Pareto optimization in combining multiple supervision sources, representing a novel framework compared to previous weak supervision work.
2. The Pareto optimal learning assessed risk (POLAR) score from our framework is shown to be effective in estimating LLM error probability.
3. A dynamic prompting strategy designed to automatically improve high-risk instances outperforms SOTA supervised models without any manually labeled training data.

## 2 RELATED WORK

Early works in model calibration date back to the seminal work of Platt scaling (Platt et al., 1999), where a Logistic calibration model is fitted on top of the original model output. Various techniques have been developed afterwards for model calibration, including isotonic regression (Zadrozny & Elkan, 2002), temperature scaling (Guo et al., 2017), and Bayesian binning (Naeini et al., 2015). A

contextual calibration method for LLMs was proposed by Zhao et al. (2021), which adjust the class balance by taking an ensemble of LLM queries with content-free input. Most of these methods rely on labeled calibration data, and there isn't a systematic self-supervised approach to our knowledge.

As accurate calibration for LLM is challenging, heuristic methods have been proposed to detect hallucination or estimate the confidence level of the answer. Wang et al. (2022) used self-consistency to infer the reliability of the answer. Manakul et al. (2023) proposed SelfCheckGPT as a black-box method to detect hallucination. The chain-of-thought (CoT) by Wei et al. (2022) has also been used to indicate potential errors. These methods are less quantitative and are vulnerable that the estimated confidence is biased by the model itself. The quality of the results could be highly dependent on the prompting strategy, and there is no systematical way to quantify this.

Our work steams from the weak supervision domain that aggregates multiple supervision sources in a single label (Zhang et al., 2022). Following early works in distant supervision (Hoffmann et al., 2011) and crowd-sourcing (Raykar et al., 2010), data programming was proposed by Ratner et al. (2016; 2017) as one of the milestones. Numerous methods have been introduced into the field (Fu et al., 2020; Varma et al., 2019; Wang & Poon, 2018; Lang & Poon, 2021), with MeTaL (currently known as Snorkel) (Ratner et al., 2019) as the most popular method as of today. Most of the existing works weight the sources across all examples, leading to the weighting dilemma. Rühling Cachay et al. (2021) mitigates the problem with instance-dependent weighting, but introduces more variance in learning the weights. We address this problem with Pareto optimization to be adaptive to all sources simultaneously and show that this approach offers better LLM calibration ability.

## 3 METHODOLOGY

### 3.1 PROBLEM SETUP

Denote the LLM as a function $\text{LLM}(x; \text{prompt})$, parameterized by a user-defined prompt for a specific task. The LLM is required to give a correct response to the given input $x \in \mathcal{X}$. Given the challenge of evaluating LLM output in free text form (Ji et al., 2023), this study focuses on tasks where the desired output space $\mathcal{Y}$ is finite. We fix the prompt for the specific task and map LLM responses into the target space with operator $\mathcal{P}$ (e.g., taking the first token), defining

$$\Lambda(x) := \mathcal{P}(\text{LLM}(x; \text{prompt})) \in \mathcal{Y} \cup \{0\}, \tag{1}$$

where 0 signifies the LLM stating "unsure" when lacking confidence in its answer. The error or hallucination is defined as the model confidently stating an answer ($\Lambda(x) \neq 0$) that is actually incorrect ($\Lambda(x) \neq y$). The goal is to estimate $\mathbb{P}(\Lambda(x) \neq y | \Lambda(x) \neq 0)$. Our self-supervision framework utilizes the following components:

- Unlabeled input examples $x_1, \cdots, x_n \in \mathcal{X}$.
- LLM function $\Lambda(x) := \mathcal{P}(\text{LLM}(x; \text{prompt})) \in \mathcal{Y} \cup \{0\}$.
- $m$ *Supervision Functions*[1] that can be triggered to output heuristic labels:

$$\lambda_j(x) \in \mathcal{Y} \cup \{0\}, \ j = 1, \cdots, m. \tag{2}$$

The supervision functions, $\lambda_j(x)$, are heuristic rules based on elements like keywords, regular expressions, or existing knowledge bases. Again, 0 refers to an *abstain* output where the supervision function is not triggered. The supervision function $\lambda_j$'s are expected to be better than a random guess (Ratner et al., 2016; Wu et al.), as detailed in Section 3.2.

### 3.2 HOW TO CALIBRATE?

In Section 1, we described two intuitions for achieving calibration through self-supervision. Here, we formally state corresponding propositions, with detailed proof provided in Appendix A. To clarify, these propositions do not serve as the "lemmas" for our main result in the next section, but are meant for inspiration for future work.

---

[1]Referred to as *labeling functions* in some literature Zhang et al. (2022). We use the term supervision function for more generalized settings.

**Proposition 1.** *Suppose the ground truth label $y \in \mathcal{Y}$ is unseen, and an imperfect teacher model $\Lambda(x)$ has an error rate $\alpha$ with miss-classification evenly distributed:*

$$\mathbb{P}(\Lambda(x) = y) = 1 - \alpha, \quad \mathbb{P}(\Lambda(x) = y') = \frac{\alpha}{|\mathcal{Y}| - 1}, \forall\, y' \neq y,\, \forall x \in \mathcal{X}. \tag{3}$$

*Then fitting a student model $h(x)$ to $\Lambda(x)$ is equivalent to training on ground truth labels with a label smoothing parameter $\alpha$ as defined in Section 1.1 by Müller et al. (2019).*

We leave the straightforward proof to Appendix A. Müller et al. (2019) empirically demonstrated that label smoothing improves or at least doesn't harm model calibration compared to standard supervised learning. Therefore, Proposition 1 suggests that access to ground truth labels is not necessary for obtaining well-calibrated outputs. If LLM errors were uniformly "random", we could directly use it as a teacher model to distill a small network on the specific task for calibration.

However, LLM errors are rarely "random". Deterministic functions typically err on specific types of inputs. Distilling a model solely from the LLM could lead to overfitting, thus unable to signal errors in the teacher LLM. Our solution is to integrate other weak supervision functions as external signals independent of the LLM. As each supervision source makes different types of errors on different examples, their ensemble approximates a "random" scenario. Concerns that lower quality external sources might undermine the LLM's high performance are addressed in our next proposition.

**Proposition 2.** *Consider target space $\mathcal{Y} = \{-1, 1\}$ for simplicity. Suppose the LLM $\Lambda(x)$ is arbitrarily accurate with $\mathbb{P}(\Lambda(x) = y) = p < 1$. Model the noisy weak signals ensemble as $w(x) \sim \mathcal{N}(y \cdot \mu, \sigma^2)$ with $\mu > 0$. If $w(x) \perp \Lambda(x) \mid y$, a function $\psi(\Lambda(x), w(x))$ always exists s.t.*

$$\mathbb{P}(\psi(\Lambda(x), w(x)) = y) > p.$$

We prove by parameterization in Appendix A. Here the weak signal ensemble $w(x)$ is a continuous Gaussian r.v. to model the scenario where many independent signals are aggregated together. The implication of this proposition is that even for an LLM with high accuracy, any weak supervision signal ensemble can enhance or at least not reduce the LLM's performance, provided they are slightly better than a random guess and independent of the LLM conditional on the true label.

### 3.3 PARETO OPTIMAL LEARNING

In this section we take a theoretical leap from the previous inspirational propositions and state our main results. For any specific task, we utilize the LLM $\Lambda(x)$ and $m$ heuristic supervision functions $\lambda_j(x)$ to auto-generate noisy labels. We then fit a smaller network to all imperfect supervision sources simultaneously, defining it as the *harmonizer* model $h : \mathcal{X} \to \mathcal{Y}$. Incorporating multiple sources helps reducing labeling error dependency. To this point, the primary challenge is designing a framework to resolve conflicts between the LLM and weak sources.

Contrary to the often-assumed independence of weak sources in previous literature, we adopt a looser assumption: the LLM and each weak source are individually positively correlated with the ground truth (i.e., better than a random guess). Consequently, a reasonable harmonizer model $h(x)$ should align with each source. Mathematically, it solves the following multi-objective problem:

$$\min_{h \in \mathcal{H}} \quad \mathbb{E}x \sim \mathcal{X}[\ell_0(h(x), \Lambda(x))], \quad \mathbb{E}x \sim \mathcal{X}[\ell_j(h(x), \lambda_j(x))]_{j=1}^m. \tag{4}$$

As the objectives may conflict, we seek a harmonizer $h^* \in \mathcal{H}$ that is *Pareto optimal*, following multi-objective learning theory (Hwang & Masud, 2012) and Pareto optimization (Pareto, 1964).

**Definition 1** (Pareto optimal harmonizer). $h^* \in \mathcal{H}$ is a Pareto optimal harmonizer to $\lambda_0, \cdots, \lambda_m$, if no $h \in \mathcal{H}$ exists that Pareto dominates $h^*$ in Problem 4. Mathematically, $h^*$ must satisfy

$$\nexists h \in \mathcal{H}, \quad s.t. \begin{cases} \forall j = 0, 1, \cdots, m, & \mathbb{E}[\ell_j(h(x), \lambda_j(x))] \leq \mathbb{E}[\ell_j(h^*(x), \lambda_j(x))], \\ \exists j = 0, 1, \cdots, m, & \mathbb{E}[\ell_j(h(x), \lambda_j(x))] < \mathbb{E}[\ell_j(h^*(x), \lambda_j(x))]. \end{cases}$$

Here, $\lambda_0 := \Lambda$ is used for simplicity. The Pareto optimization framework effectively manages dependencies between supervision sources, for example, a harmonizer's Pareto optimality remains unaffected by the arbitrary duplication of supervision functions. However, finding Pareto optimal

solutions remains challenging in the multi-objective optimization literature. Our approach scalarizes the multiple objectives by minimize the following Pareto loss scalarizer $G : \mathbb{R}_+^{m+1} \to \mathbb{R}_+$:

$$\min_{h \in \mathcal{H}} \quad \mathbb{E}_{x \sim \mathcal{X}}[G\left(\ell_0(h(x), \Lambda(x)), \ \ell_1(h(x), \lambda_1(x)), \ \cdots, \ \ell_m(h(x), \lambda_m(x))\right)]. \tag{5}$$

$G : \mathbb{R}+^{m+1} \to \mathbb{R}+$ must meet the following conditions:

**Definition 2** (Pareto scalarizer). $G(\ell_0, \ell_1, \cdots, \ell_m)$ is a *Pareto scalarizer*, if it satisfies:

- $G(\ell_0, \cdots, \ell_j', \cdots, \ell_m) < G(\ell_0, \cdots, \ell_j, \cdots, \ell_m)$ if $\ell_j' < \ell_j$, for $\forall j = 0, 1, \cdots, m$;

- $G : \mathbb{R}_+^{m+1} \to \mathbb{R}_+$ is convex.

In this study, we explore four different types of scalarizers:

- Linear scalarizer: $G(\ell_0, \ell_1, \cdots, \ell_m) = \sum_{j=0}^m \ell_j = \|\vec{\ell}\|_1$.

- Quadratic scalarizer: $G(\ell_0, \ell_1, \cdots, \ell_m) = \left(\sum_{j=0}^m \ell_j\right)^2 = \|\vec{\ell}\|_1^2$.

- Euclidean norm scalarizer: $G(\ell_0, \ell_1, \cdots, \ell_m) = \sqrt{\sum_{j=0}^m \ell_j^2} = \|\vec{\ell}\|_2$.

- Chebyshev scalarizer: $G(\ell_0, \ell_1, \cdots, \ell_m) = \max_{j=0}^m \ell_j = \|\vec{\ell}\|_\infty$.

The nonlinear function of $\vec{\ell}$ shapes the optimization in Eq. 5 differently through Jensen's inequality. For example, the quadratic scalarizer places more emphasis on challenging examples with large overall losses. While the first three scalarizers qualify as Pareto scalarizers, the Chebyshev scalarizer does not meet the definition criteria, serving as a comparative element in our experiment.

For any Pareto scalarizer, the proposed approach in Eq. 5 is guaranteed by the following theorem.

**Theorem 1.** *Suppose $G : \mathbb{R}_+^{m+1} \to \mathbb{R}_+$ is a Pareto scalarizer as in Definition 2. Solving the problem in Equation 5 approximates a Pareto optimal harmonizer by minimizing the upperbound.*

We refer to Appendix A for a detailed proof of the theorem. According to Theorem 1, any Pareto scalarizer defined in Definition 2 can approximate a Pareto optimal harmonizer by minimizing the upper bound. Based on the theoretical analysis, we propose identifying the harmonizer model $h(x)$ for LLM calibration by solving problem 5 using stochastic gradient-based algorithms such as Adam (Kingma & Ba, 2014). To ensure high-quality calibration from $h(x)$, we recommend using cross-entropy loss for $\ell_0, \ell_1, \cdots, \ell_m$. Once an optimal solution $h^* \in \mathcal{H}$ is found, for any input $x$ and LLM response $\Lambda(x)$, the Pareto optimal learning assessed risk (POLAR) score is defined as

$$\zeta(x, \Lambda(x); h^*) = \mathbb{P}_{Y \sim h^*(x)}(\Lambda(x) \neq Y | \Lambda(x) \neq 0), \tag{6}$$

where $\mathbb{P}_{Y \sim h^*(x)}$ represents the probability distribution of $Y$ as estimated by $h^*(x)$. Algorithm 1 summarizes the entire process.

---

**Algorithm 1** POLAR for LLM responses

---

1: **Input:** LLM response $\Lambda$ as in Equation 1, supervision functions $\lambda_1, \cdots, \lambda_m$ as in Equation 2, unlabeled training examples $x_{1:n}$. Initialize harmonizer $h \in \mathcal{H}$.
2: **for** $i = 1$ to $n$ **do**
3:     $\text{loss}_{\text{LLM}} = (\Lambda(x_i) \neq 0) * \ell(h(x_i), \ \Lambda(x_i))$
4:     **for** $j = 1$ to $m$ **do**
5:         $\text{loss}_j = (\lambda_j(x_i) \neq 0) * \ell(h(x_i), \ \lambda_j(x_i))$
6:     **end for**
7:     Update $h$ with SGD iteration of $\min_h G(\text{loss}_{\text{LLM}}, \ \text{loss}_1, \cdots, \text{loss}_m)$.
8: **end for**
9: **Output:** Harmonizer $h^*$. For any example $x$ and LLM response $\Lambda(x)$, the POLAR score of the response is estimated as according to Equation 6.

---

Let's make a brief theoretical summary here. If the LLM makes uniformly "random" errors, simply distilling a smaller network would give a calibration model. Since this is not usually the case, we

incorporate multiple independent supervision functions alongside the LLM, fitting a harmonizer model to all simultaneously to prevent overfitting to the LLM. Theoretically, we approximate a harmonizer model that is Pareto optimal. However, due to the complexity of modeling the errors made by the LLM and supervision functions, we cannot offer further theoretical guarantees at this point. We empirically demonstrate the calibration capability of the harmonizer model in Section 4.

### 3.4 LLM ERROR CORRECTION WITH POLAR-ASSISTED DYNAMIC PROMPTING

Identifying LLM responses with a high risk of error presents an opportunity to automatically improve these responses. To this end, we propose two dynamic prompting strategies, both supported by the POLAR score, to illustrate the POLAR score's potential in correcting LLM errors:

**Dynamic self-examination**  In this strategy, whenever the POLAR score $\zeta(x, \Lambda(x); h^*) > \delta$ for threshold $\delta$, we ask the LLM to reflect on its answer. See appendix for detailed prompt.

**Dynamic self-supervision**  In this strategy, we utilize the supervision functions as sources to help the LLM reflect on its initial response (e.g. stating the reg-ex of knowledge base findings).

Algorithm 2 outlines the POLAR-assisted dynamic prompting strategy. We provide detailed prompting design description in the Appendix D.

---

**Algorithm 2** POLAR-assisted dynamic self-supervision for LLM

---

1: **Input:** Example $x$, LLM response $\Lambda(x)$, supervision functions $\lambda_1, \cdots, \lambda_m$, harmonizer $h^*$.
2: **if** $\zeta(x, \Lambda(x); h^*) > \delta$ **then**
3:     Initialize *Reflection Prompt*.
4:     **for** $j = 1$ to $m$ **do**
5:         **if** $\lambda_j(x) \neq 0$ **then**
6:             Add evidence from supervision function $j$ to the *Reflection Prompt*.
7:         **end if**
8:     **end for**
9:     Respond to the LLM with the *Reflection Prompt* and get new response $\Lambda'(x)$.
10:     **return** $\Lambda'(x)$
11: **end if**

---

## 4 EXPERIMENTS

**Dataset**  Since supervision functions are crucial in our framework, for reproducibility, we focus exclusively on tasks with publicly available supervision functions, primarily developed in the weak supervision literature. We reference benchmark datasets collected by Zhang et al. (2021) and evaluate on four different NLP tasks: CDR (Li et al., 2016), ChemProt (Krallinger et al., 2017), SemEval (Hendrickx et al., 2019), and SMS (Almeida et al., 2011). Training set labels are removed, while gold labels are retained on test set for evaluation. Our dataset selection covers the following aspects:

- Domain: General domain (SemEval, SMS) and biomedical domain (CDR, ChemProt).
- Task: Relation extraction (CDR, ChemProt, SemEval) and classification (SMS).
- Difficulty: Tasks easily solvable by advanced LLMs like GPT-4, e.g., SMS (99% F-1), and those still challenging (CDR (74% F-1), ChemProt (42% F-1), and SemEval (67% F-1)).

**Prompt design**  To maximize LLM capabilities, we carefully design prompts for each problem, clarifying the problem setting, knowledge background, input/output structure, and instructions for stating "unsure". Detailed prompt information is provided in the appendix.

**Supervision functions**  The supervision functions are based on simple rules set by human experts (Ratner et al., 2017; Yu et al., 2021; Zhou et al., 2020; Awasthi et al., 2020), including:

- Keywords and regular expression pattern checking.

- Knowledge base (e.g. the Comparative Toxicogenomics Database (Davis et al., 2021))
- Hierarchical combination of the above methods.

**Harmonizer training** Our harmonizer $h(x)$ is trained through Pareto optimal learning, as per Equation 5 and Algorithm 1. The primary experiments employ BERT (Devlin et al., 2018) (and PubMedBERT (Gu et al., 2020) for biomedical datasets CDR and ChemProt) as the harmonizer model, with a quadratic scalarizer. Alternative approaches are discussed in Section 5, with training specifics in Appendix C.

## 4.1 POLAR CALIBRATION OF LLM ERROR

In this section we present our implementation results of Algorithm 1. The fitted harmonizer is applied to unseen test set to give a POLAR score for each LLM response. Gold labels on the test set are used to estimate the actual error rate, evaluating the POLAR score's calibration accuracy.

Figure 2 displays POLAR score calibration for GPT-4 on the CDR chemical-disease relation extraction task. The calibration curve (Figure 2a) confirms that the POLAR score reliably estimates the true probability of LLM errors. Figure 2b demonstrates a high correlation between the POLAR score and error rate. Lastly, Figure 2c reveals that responses with the highest POLAR scores are most prone to hallucinations or errors, with the top scores indicating nearly a 100% error rate.

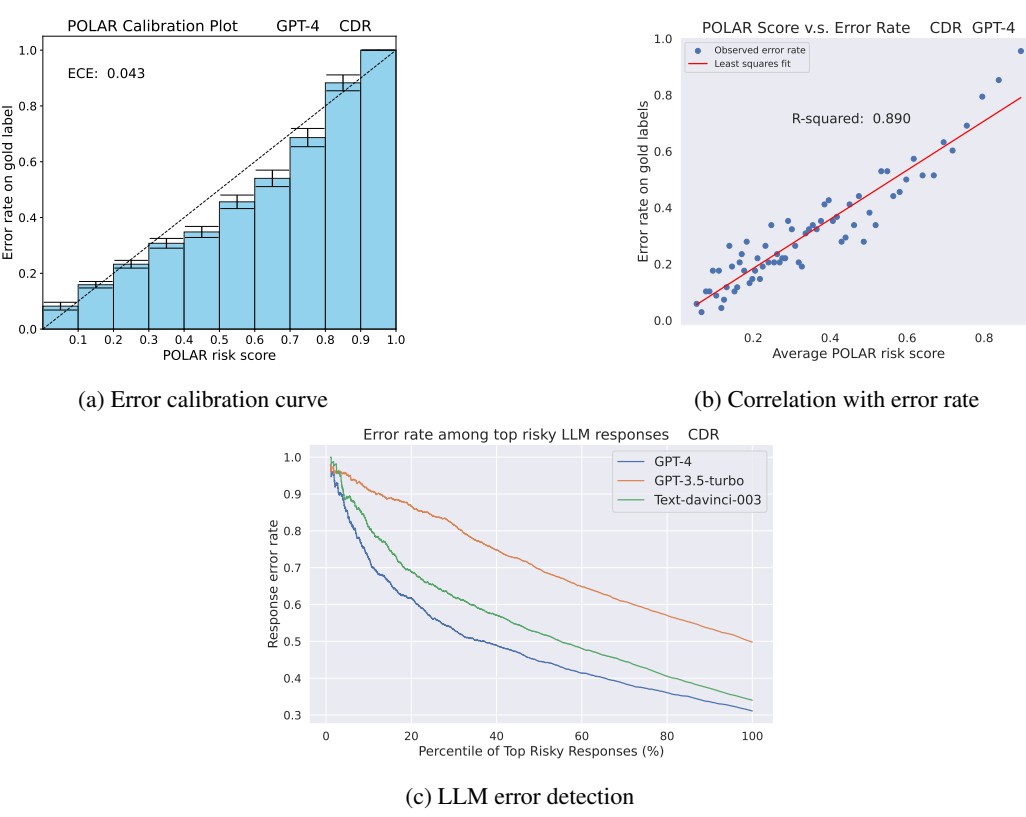

(a) Error calibration curve

(b) Correlation with error rate

(c) LLM error detection

Figure 2: LLM error calibration and hallucination detection using POLAR score. Figure 2a shows the error rate on the test set for LLM responses ranked into ten POLAR score bins. The expected calibration error (ECE) is the weighted average of the absolute deviation of POLAR score from the actual error rate in each bin. Figure 2b ranks the LLM responses into POLAR score bins each with 100 examples, and plot the average POLAR score and error rate for each bin. The $R^2$ is calculated on the scatter plot. Figure 2c shows the LLM error rate among the top POLAR score examples.

Table 1 compares POLAR score performance in LLM error calibration against baseline methods, using metrics such as Expected Calibration Error (ECE) and $R^2$ between estimated error rate and true

error rate (details in Figure 2). We report the results spanning four datasets and three LLMs (GPT-4, GPT-3.5-turbo, and text-davinci-003). The following four methods serve as baseline comparison:

- Snorkel (Ratner et al., 2019): One of the most recommended weak supervision method (Zhang et al., 2021) combining multiple supervision sources via matrix completion. Snorkel model fitted on training set is use to give class probabilities for LLM error rate.

- WeaSEL (Rühling Cachay et al., 2021): One of the state of the art weak supervision framework that fits a neural network using the LLM and weak labels in an end-to-end manner. We use the class probability given by the fitted end model to estimate LLM error rate.

- Majority vote: Another popular weak supervision method that estimates class probability according to the voted ratios among the LLM and the supervision functions.

- LLM distilled: Fit a BERT (PubMedBERT) model to the LLM output on the task directly. Take the class probabilities from the finetuned BERT model to estimate LLM error rate.

- LLM ensemble: Query the LLM multiple times and estimate class probability from the response ensemble. As the approach is extremely expensive, we only implement for GPT-3.5-turbo on CDR with resulting ECE = 0.4466, far from comparable to other approaches.

As shown in Table 1, for all three LLMs across all four tasks, the proposed POLAR score consistently outperforms other methods. Among the baseline methods, Snorkel, WeaSEL, and LLM distilled model can achieve top or close-to-top performance in some cases under specific metric, but lack the consistency to deliver stable calibration for different LLMs on different tasks. In comparison, the proposed POLAR score is consistently well-calibrated to the true error rate.

Table 1: LLM error calibration using POLAR score, compared with baseline methods. The best entries from each row in terms of low ECE and high $R^2$ are highlighted in bold.

| | POLAR | | Snorkel | | WeaSEL | | Majority vote | | LLM Distilled | |
|---|---|---|---|---|---|---|---|---|---|---|
| **CDR** | ECE | $R^2$ | ECE | $R^2$ | ECE | $R^2$ | ECE | $R^2$ | ECE | $R^2$ |
| GPT-4 | **0.043** | **0.890** | 0.167 | 0.299 | 0.146 | 0.387 | 0.145 | 0.348 | 0.164 | 0.592 |
| GPT-3.5-turbo | **0.046** | 0.934 | 0.164 | 0.320 | 0.081 | **0.942** | 0.182 | 0.540 | 0.222 | 0.037 |
| Text-davinci-3 | **0.055** | **0.907** | 0.154 | 0.371 | 0.135 | 0.877 | 0.149 | 0.450 | 0.089 | 0.896 |
| **ChemProt** | ECE | $R^2$ | ECE | $R^2$ | ECE | $R^2$ | ECE | $R^2$ | ECE | $R^2$ |
| GPT-4 | **0.035** | **0.934** | 0.182 | 0.510 | 0.278 | 0.885 | 0.233 | 0.244 | 0.216 | 0.766 |
| GPT-3.5-turbo | **0.048** | **0.944** | 0.228 | 0.625 | 0.219 | 0.922 | 0.282 | 0.031 | 0.159 | 0.845 |
| Text-davinci-3 | **0.051** | **0.917** | 0.218 | 0.700 | 0.213 | 0.846 | 0.279 | 0.307 | 0.196 | 0.825 |
| **SemEval** | ECE | $R^2$ | ECE | $R^2$ | ECE | $R^2$ | ECE | $R^2$ | ECE | $R^2$ |
| GPT-4 | 0.079 | 0.916 | 0.068 | 0.714 | 0.612 | 0.626 | 0.115 | 0.379 | **0.063** | **0.947** |
| GPT-3.5-turbo | **0.047** | **0.963** | 0.150 | 0.821 | 0.345 | 0.890 | 0.277 | 0.208 | 0.108 | 0.757 |
| Text-davinci-3 | 0.067 | **0.950** | 0.119 | 0.796 | 0.455 | 0.784 | 0.242 | 0.396 | **0.065** | 0.936 |
| **SMS** | ECE | $R^2$ | ECE | $R^2$ | ECE | $R^2$ | ECE | $R^2$ | ECE | $R^2$ |
| GPT-4 | 0.014 | **0.980** | 0.244 | 0.089 | 0.409 | 0.345 | 0.588 | 0.091 | **0.013** | 0.977 |
| GPT-3.5-turbo | **0.041** | **0.963** | 0.075 | 0.202 | 0.286 | 0.731 | 0.148 | 0.006 | 0.131 | 0.775 |
| Text-davinci-3 | **0.023** | 0.943 | 0.201 | 0.053 | 0.420 | 0.238 | 0.325 | 0.091 | 0.033 | **0.956** |

## 4.2 LLM ERROR CORRECTION WITH POLAR-ASSISTED DYNAMIC PROMPTING

This experiment explores dynamic prompting strategies from Section 3.4 to rectify LLM errors. As an extension experiment, we only focus on the CDR dataset for GPT-4 and GPT-3.5-turbo. We sort the initial LLM responses by their POLAR score, and compute the error rate before and after dynamic prompting. Figure 3a shows that the GPT-4 error rate decreases significantly for both strategies, if and only if the POLAR score was high. Otherwise, re-prompting for self-examine or self-supervision can even increase error rate. Therefore, it is essential to perform dynamic prompting with the POLAR score information.

Figure 3b presents results from dynamic self-examination and self-supervision as per Algorithm 2. With a set POLAR score threshold $\delta = 0.5$, both strategies improve GPT-3.5-turbo and GPT-4 performances. Notably, dynamic self-supervision allows GPT-3.5-turbo to surpass previous best results without gold-labeled data Zhang et al. (2021) and enables GPT-4 to outperform state-of-the-art supervised methods Xiao et al. (2021), all without utilizing any gold-labeled examples.

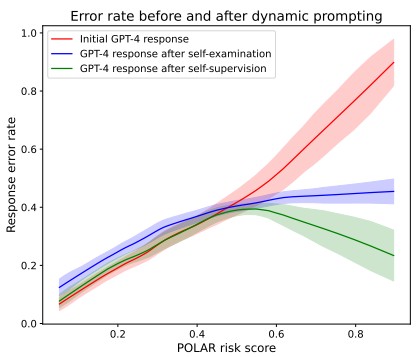

(a) POLAR score and LLM error rate change

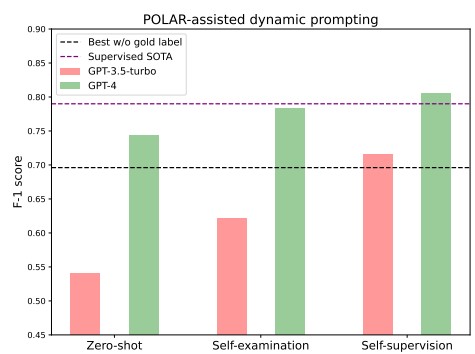

(b) Dynamic prompting performance

Figure 3: (a) GPT-4 error rate before and after dynamic prompting, conditional on the initial POLAR score. (b) shows the performance improvement using the two dynamic prompting strategies.

## 5 COMPONENTS FOR CALIBRATION

**External sources of information** The supervision functions act as external information sources, aiding in the harmonizer's calibration and preventing overfitting to LLM responses. To illustrate the importance of these external sources, we examine the performance of models without them. As shown in the LLM distilled model columns in Table 1, the absence of supervision functions generally leads to inconsistent calibration capabilities, primarily due to overfitting to LLM outputs.

**Pareto loss scalarizer and harmonizer** The effectiveness of different loss scalarizers and harmonizer models is encapsulated in Table 2, where we present the Expected Calibration Error (ECE) and $R^2$ measures averaged across three LLMs for the four tasks. We observe that the nonlinear quadratic loss scalarizer, when combined with BERT finetuning, delivers superior calibration performance. Conversely, simpler models, such as Multi-Layer Perceptron (MLP) and Logistic Regression (LR), achieve optimal results with a basic linear scalarizer. Notably, the Chebyshev scalarizer consistently underperforms across various scenarios. This empirical evidence lends support to Theorem 1, validating the necessity of a Pareto loss scalarizer for approximating a Pareto optimal harmonizer.

Table 2: Average calibration ability for different loss scalarizer $G$ and harmonizer type.

| $G$ function | Linear | | Quadratic | | Euclidean norm | | Chebyshev | |
|---|---|---|---|---|---|---|---|---|
| Harmonizer type | ECE | $R^2$ | ECE | $R^2$ | ECE | $R^2$ | ECE | $R^2$ |
| BERT | 0.0625 | 0.9273 | **0.0458** | **0.9366** | 0.0549 | 0.9003 | 0.0711 | 0.8260 |
| MLP | **0.0555** | **0.9392** | 0.0974 | 0.9188 | 0.0691 | 0.9302 | 0.0775 | 0.8934 |
| LR | **0.0641** | **0.9360** | 0.1072 | 0.9020 | 0.0766 | 0.9288 | 0.0948 | 0.8813 |

## 6 CONCLUSION

We present a novel framework for LLM calibration using Pareto optimal self-supervision. Our key innovation, the POLAR (Pareto Optimal Learning Assessed Risk) score, aligns well with LLM error probabilities, ensuring reliable calibration. Significantly, our POLAR-based dynamic prompting strategy enhances GPT-4's performance, surpassing state-of-the-art supervised models without the need for human-labeled training data. This development marks a substantial advancement in LLM application and opens new directions in self-supervised learning within artificial intelligence.

## REPRODUCIBILITY STATEMENT

The proof of the theoretical results are in Appendix A. Implementation of our experiments are illustrated in Algorithms 1 and 2. The training details of the harmonizer model are listed in Appendix C. The prompts for querying the LLMs are described in Appendix D to reproduce the response. Anonymized code is provided in the supplementary material.

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

## A    PROOF OF THEOREMS

**Proposition A.1.** *Suppose the ground truth label $y \in \mathcal{Y}$ is unseen, and the imperfect supervision model $\Lambda(x)$ has error rate $\alpha$ with miss-classification evenly distributed:*

$$\mathbb{P}(\Lambda(x) = y) = 1 - \alpha, \quad \mathbb{P}(\Lambda(x) = y') = \frac{\alpha}{|\mathcal{Y}| - 1}, \ y' \neq y. \tag{7}$$

*Then fitting a model $h(x)$ to $\Lambda(x)$ is equivalent to training on ground truth labels with label smoothing as defined in Section 1.1 by Müller et al. (2019).*

*Proof.* Let's first vectorize $y$ and $\Lambda(x)$. Denote $y_k = 1$ if $y = k$, otherwise $y_k = 0$. Denote $\Lambda_k = 1$ if $\Lambda(x) = k$, otherwise $\Lambda_k = 0$. Let $p_k$ be the probability of $\mathbb{P}(h(x) = k)$. Fitting a model $h(x)$ to $\Lambda(x)$ is essentially minimizing

$$\mathbb{E}\left[ \sum_{k=1}^{|\mathcal{Y}|} -\Lambda_k \cdot \log(p_k) \right]$$

$$= \mathbb{E}\left[ \sum_{k=1}^{|\mathcal{Y}|} -(1-\alpha)y_k \cdot \log(p_k) - \sum_{k' \neq k} \frac{\alpha}{|\mathcal{Y}| - 1} y_{k'} \cdot \log(p_k) \right]$$

$$= \mathbb{E}\left[ \sum_{k=1}^{|\mathcal{Y}|} -(y_k(1 - \beta) + \beta/|\mathcal{Y}|) \cdot \log(p_k) \right],$$

where $\beta = \frac{|\mathcal{Y}|}{|\mathcal{Y}| - 1}\alpha$ is the label smoothing parameter. $\square$

**Proposition A.2.** *Consider target space $\mathcal{Y} = \{-1, 1\}$ for simplicity. Suppose the LLM is arbitrarily accurate with $\mathbb{P}(\Lambda(x) = y) = p < 1$, and the weak independent signal ensemble is modeled by $w(x) \sim \mathcal{N}(y \cdot \mu, \sigma^2)$ with $\mu > 0$, then there always exist a function $\psi(\Lambda, w)$ s.t.*

$$\mathbb{P}(\psi(\Lambda(x), w(x)) = y) > p.$$

*Proof.* Let's prove by constructing the function class

$$\psi(\Lambda, w) = \text{sign}(\alpha\Lambda + w).$$

Suppose $y = 1$ WLOG. Then the accuracy is

$$\mathbb{P}(\alpha\Lambda + w > 0) = p \cdot \Phi\left(\frac{\mu + \alpha}{\sigma}\right) + (1 - p) \cdot \Phi\left(\frac{\mu - \alpha}{\sigma}\right),$$

where $\Phi(\cdot)$ is the normal distribution CDF.

As $\alpha \to \infty$, $\mathbb{P}(\alpha\Lambda + w > 0) \to p$ as it is equivalent to the LLM itself. We just need to show that the $\frac{\partial \mathbb{P}(\alpha\Lambda + w > 0)}{\partial \alpha} < 0$ as $\alpha \to \infty$. Let's write down

$$\frac{\partial \mathbb{P}(\alpha\Lambda + w > 0)}{\partial \alpha} = \frac{p}{\sigma} \cdot \phi\left(\frac{\mu + \alpha}{\sigma}\right) - \frac{1 - p}{\sigma} \cdot \phi\left(\frac{\mu - \alpha}{\sigma}\right).$$

As both terms are positive, the condition now is for

$$\frac{p}{1 - p} \cdot \phi\left(\frac{\mu + \alpha}{\sigma}\right) / \phi\left(\frac{\mu - \alpha}{\sigma}\right) < 1$$

Writing down the normal PDF $\phi(\cdot)$, we just need

$$\exp\left(\frac{(\mu + \alpha)^2 - (\mu - \alpha)^2}{2\sigma^2}\right) = \exp\left(\frac{2\mu\alpha}{\sigma^2}\right) > \sqrt{2\pi}\frac{p}{1 - p}.$$

which is easily achieved by the exponential function as $\alpha \to \infty$. $\square$

**Definition A.1** (Pareto optimal harmonizer). $h^* \in \mathcal{H}$ is a Pareto optimal harmonizer to $\Lambda$ and $\lambda_1, \cdots, \lambda_m$, if there does not exist any $h \in \mathcal{H}$ that Pareto dominates $h^*$ in Problem **??**. Mathematically, if we denote $\lambda_0 := \Lambda$, $h^*$ needs to satisfies the following:

$$\nexists h \in \mathcal{H}, \quad s.t. \begin{cases} \forall j = 0, 1, \cdots, m, & \mathbb{E}[\ell_j(h(x), \lambda_j(x))] \leq \mathbb{E}[\ell_j(h^*(x), \lambda_j(x))], \\ \exists j = 0, 1, \cdots, m, & \mathbb{E}[\ell_j(h(x), \lambda_j(x))] < \mathbb{E}[\ell_j(h^*(x), \lambda_j(x))]. \end{cases}$$

For our specific problem, we propose to approximate the problem by minimize the following Pareto loss function $G : \mathbb{R}_+^{m+1} \to \mathbb{R}_+$:

$$\min_{h \in \mathcal{H}} \quad \mathbb{E}_{x \sim \mathcal{X}}[G(\ell_0(h(x), \Lambda(x)), \ell_1(h(x), \lambda_1(x)), \cdots, \ell_m(h(x), \lambda_m(x)))]. \tag{8}$$

We require $G : \mathbb{R}_+^{m+1} \to \mathbb{R}_+$ to satisfy the following conditions.

**Definition A.2** (Pareto scalarizer). $G(\ell_0, \ell_1, \cdots, \ell_m)$ is a *Pareto scalarizer*, if it satisfies:

- $G(\ell_0, \cdots, \ell_j', \cdots, \ell_m) < G(\ell_0, \cdots, \ell_j, \cdots, \ell_m)$ if $\ell_j' < \ell_j$, for $\forall j = 0, 1, \cdots, m$;

- $G : \mathbb{R}_+^{m+1} \to \mathbb{R}_+$ is convex.

**Theorem A.1.** *Suppose $G : \mathbb{R}_+^{m+1} \to \mathbb{R}_+$ is a Pareto scalarizer as in Definition A.2. Solving the problem in Equation 8 approximate a Pareto optimal harmonizer by upperbounding an objective whose optimal solution is Pareto optimal as in Definition A.1.*

*Proof.* For convenience, let's denote

$$u_j(h) := \mathbb{E}[\ell_j(h(x), \lambda_j(x))], \quad j = 0, 1, \cdots, m.$$

We first show that any $h^*$ minimizing $G(u_0, u_1, \cdots, u_m)$ is Pareto optimal.

Proof by contradiction. Suppose $h^*$ is not Pareto optimal. Then there must exist some $h' \in \mathcal{H}$ Pareto dominating $h^*$. Without loss of generality, let's assume $u_j(h') < u_j(h^*)$, and $u_k(h') \leq u_k(h^*)$, $\forall k \neq j$. Then according to Definition A.2 of Pareto scalarizer,

$$G(u_0(h'), \cdots, u_j(h'), \cdots, u_m(h')) \tag{9}$$
$$\leq G(u_0(h^*), \cdots, u_j(h'), \cdots, u_m(h^*)) \tag{10}$$
$$< G(u_0(h^*), \cdots, u_j(h^*), \cdots, u_m(h^*)), \tag{11}$$

which contradicts the assumption that $h^*$ is the minimizer for

$$G(u_0(h), \cdots, u_j(h), \cdots, u_m(h)).$$

Therefore, the original statement is true, and minimizing the objective

$$\min_{h \in \mathcal{H}} \quad G(\mathbb{E}[\ell_0(h(x), \Lambda(x))], \mathbb{E}[\ell_1(h(x), \lambda_1(x))], \cdots, \mathbb{E}[\ell_m(h(x), \lambda_m(x))]) \tag{12}$$

gives a Pareto optimal harmonizer.

Next, we use Jensen's inequality to upperbound this objective with the objective in problem 8. Using the fact that $G$ is convex, we apply Jensen's inequality and get

$$G(\mathbb{E}[\ell_0(h(x), \Lambda(x))], \mathbb{E}[\ell_1(h(x), \lambda_1(x))], \cdots, \mathbb{E}[\ell_m(h(x), \lambda_m(x))]) \tag{13}$$
$$\leq \quad \mathbb{E}_{x \sim \mathcal{X}}[G(\ell_0(h(x), \Lambda(x)), \ell_1(h(x), \lambda_1(x)), \cdots, \ell_m(h(x), \lambda_m(x)))]. \tag{14}$$

Therefore, solving the problem in Equation 8 approximates Pareto optimal harmonizer by upperbounding Equation 12. □

## B   WEIGHTS FOR REBALANCING THE SOURCES

In our experiments, we explored four different types of scalarization functions, namely:

- Linear scalarizer: $G(\ell_0, \ell_1, \cdots, \ell_m) := \sum_{j=0}^{m} w_j \ell_j$.

- Quadratic scalarizer: $G(\ell_0, \ell_1, \cdots, \ell_m) := \left( \sum_{j=0}^{m} w_j \ell_j \right)^2$.

- Euclidean norm scalarizer: $G(\ell_0, \ell_1, \cdots, \ell_m) := \| (w_0 \ell_0, w_1 \ell_1, \cdots, w_m \ell_m) \|$.

- Chebyshev scalarizer: $G(\ell_0, \ell_1, \cdots, \ell_m) := \max_{j=0}^{m} w_j \ell_j$.

The weights $w_j \in \mathbb{R}$ are parameters of $G$. In the main text of the paper, we fixed to equal weights $\vec{w} = \vec{1}$. Here we introduce three approaches to determine the weighting if necessary.

**Equal Weight**   The simplest weighting scheme of

$$w_0 = w_1 = \cdots = w_m = \frac{1}{m+1}$$

gives nice performance in practice, and is the method we used for the results in the main body of the paper. The nonlinear Pareto scalarizers have the ability to balance the sources even under equal weights. It is always recommended to start with equal weight.

In the case that the supervision sources are highly correlated, or when the quality of the sources varies a lot, we propose the following two approaches utilizing the correlation in prediction residual.

**Maximal Eigenvalue of Residual Correlation**   Suppose we have a pilot harmonizer $h_0 \in \mathcal{H}$, which can usually be obtained from minimizing a Pareto scalarizer with equal weight, it gives predicted distribution $\vec{p}(x) \in \mathbb{R}^{|\mathcal{Y}|}$ for any input $x \in \mathcal{X}$, where

$$p_c(x) := \mathbb{P}(h_0(x) = c).$$

For any source $0 \leq j \leq m$, denote the one-hot vector $\vec{\lambda}_j(x) \in \mathbb{R}^{|\mathcal{Y}|}$ as:

$$\lambda_{j,c}(x) = \begin{cases} 1 & \text{if } \lambda_j(x) = c, \\ 0 & \text{otherwise.} \end{cases}$$

The prediction redisual is defined as

$$\vec{r}_j(x) := \vec{\lambda}_j(x) - \vec{p}(x),$$

which accounts for the supervision function label for $x$ that is unexplained by the harmonizer $h_0(x)$.

In order to rebalance the sources, we consider the correlation matrix $C$ between the prediction residual $\vec{r}_j$'s. Specifically, let the covariance be

$$\Sigma_{ij} := \mathbb{E}_{x \sim \mathcal{X}} [\vec{r}_i(x) \cdot \vec{r}_j(x)] - \mathbb{E}_{x \sim \mathcal{X}} [\vec{r}_i(x)] \cdot \mathbb{E}_{x \sim \mathcal{X}} [\vec{r}_j(x)].$$

The correlation variance is denoted as

$$C_{ij} = \frac{\Sigma_{ij}}{\sqrt{\Sigma_{ii} \Sigma_{jj}}}.$$

We rebalance the sources according to the eigenvector $\vec{v}_{max} \in \mathbb{R}^{m+1}$ corresponding to the largest eigenvalue of $C$. In order to get reasonable weights, we first normalize $\vec{v}_{max}$ such that the sum of the entries equals to one. Then we project $\vec{v}_{max}$ to the weights simplex with minimal values $\frac{\epsilon}{m+1}$:

$$w_{excess,j} = \left( \vec{v}_{max,j} - \frac{\epsilon}{m+1} \right)^{+}$$

$$\vec{w}_{max} = \frac{\epsilon}{m+1} + \frac{w_{excess}}{\|w_{excess}\|_1} \cdot (1 - \epsilon).$$

This projection ensures that the weight on each source is at least $\epsilon$ portion of the value from equal weight, with the minimal ratio threshold $\epsilon \in [0, 1]$.

Maximal eigenvalue method is recommended when the sources are relatively independent, and when the quality of the sources differ a lot. Intuitively, suppose two sources tends to agree with each other when they are not fitted well by the harmonizer, because there isn't intrinsic dependency between the sources, it is likely that the true label is given by the sources. Therefore, a maximal eigenvalue rebalancing scheme puts higher weights on the sources to encourage the harmonizer to fit to the unexplained examples.

**Minimal Variance of Residual Correlation**   The same as in the maximal eigenvalue method, we consider the correlation matrix $C$ between the prediction residual $\vec{r}_j$'s. Instead of finding the maximal eigenvalue of $C$, we consider solving the following minimal variance problem:

$$\min_{v} v^T C v, \quad \text{s.t. } \mathbf{1}^T v = 1.$$

This problem admits the closed form solution of

$$v_{min} = \frac{C^{-1}\mathbf{1}}{\mathbf{1}^T C^{-1}\mathbf{1}}.$$

Again, we project $\vec{v}_{min}$ to the weights simplex with minimal values $\frac{\epsilon}{m+1}$:

$$w_{excess,j} = \left(\vec{v}_{min,j} - \frac{\epsilon}{m+1}\right)^{+}$$
$$\vec{w}_{min} = \frac{\epsilon}{m+1} + \frac{w_{excess}}{\|w_{excess}\|_1} \cdot (1 - \epsilon),$$

which ensures that the weight on each source is at least $\epsilon$ portion of the value from equal weight, with the minimal ratio threshold $\epsilon \in [0, 1]$.

The minimal variance method is a classical portfolio rebalancing strategy in financial mathematics. The intuition behind the algorithm is minimizing the risk by diversification. This rebalancing scheme is useful when there are intrinsic dependency between the sources. Suppose two sources are duplicates and always tend to give the same label, their residuals should also be highly correlated. Minimal variance optimization automatically avoid putting too much weights on the duplicating sources.

While the equal weight method typically delivers good results in the simplest way, the other two rebalancing schemes are designed to address the specific concern such as source dependency and quality. It is always recommended to check against the labels on a validation set if available.

## C    Training details

We explored different configurations of Pareto optimal learning below:

- Harmonizer model: we experiment 1. BERT Devlin et al. (2018) (PubMedBERT Gu et al. (2020) for biomedical datasets CDR and ChemProt), 2. multi-layer perceptron (MLP), 3. Logistic regression (LR). The last two are built on top of the last layer embedding of the corresponding BERT model.
- Pareto loss scalarizer: we experiment all four loss scalarization functions as defined in Section 3.3, namely linear, quadratic, Euclidean norm, and Chebyshevy scalarization.
- Optimizer: We use AdamW Loshchilov & Hutter (2017) optimizer with learning rate $[10^{-4}, 10^{-5}, 10^{-6}]$, weight decay $[10^{-4}, 10^{-5}]$, batch size 16. All hyperparameters are optimized on held out dev set.
- Computation: We trained on Azure Standard NC12s v3 with 1 Nvidia V100 GPU.

## D    LLM Prompting Details

In this section we will describe the details of the prompts used to query the LLMs.

### D.1    Out-of-the-box prompt

- Setting: describe the role of the LLM in the task, and the overall problem setting.
- Background: necessary background knowledge for domain specific tasks, including information from annotation guidelines for human annotators.
- Data structure: for relation extraction tasks, explain the definition of each entity.
- Desired output: describe the list of the desired output. For each of the categories, provide explanation and optionally some examples.
- Chain of thought (CoT): instruction to encourage the LLM to think step-by-step, articulate point-by-point, and give the response in the desired structure.
- Confidence: ask the model to state "unsure" if it is not confident about the answer.
- Example: state the example and ask the model to perform the task.

Each prompt for out-of-the-box (zero-shot) prediction contains:

- A problem setting part that depends on the specific dataset.
- A response regularization part that encourages chain-of-thought (CoT) and confidence check, and specifies proper response format.
- A task instance part that contains the input instance and restates the task to perform.

**Problem setting prompt**

- CDR: "You are an intelligent assistant to extract chemical-disease relations from academic literature. Your job is to determine if in the given piece of text, the drug (entity 1) induces the disease (entity 2) or not. Negative means the drug does NOT induce the disease. Positive means the drug induces the disease. Please use your judgement to the best of your knowledge. Your answer should be classified into the following categories: [Negative, Positive]. "

- ChemProt: "You are an intelligent assistant to extract chemical-protein interaction from academic literature. Your task is to identify the chemical-protein interactions (CHEMPROT) between entity 2: Chemical Entities Mentions (CEMs) and entity 1: Gene and Protein Related Objects (named as GPRO in the instruction below) in the given piece

of text. In brief, the chemical-protein interactions include direct interactions (when a physical contact exits between a CEM and a GPRO, in most cases this GPRO being a protein or protein family and alters its function/activity) as well as indirect regulatory interactions between CEMs and GPROs (including genes, gene products (proteins, RNA), DNA/protein sequence elements and protein families, domains and complexes) that alter either the function or the quantity of the GPRO. The guidelines below provide curation rules to evaluate if the given sentence contains a description of a chemical-protein interaction; in particular, if sufficient detail/evidence is provided for comentioned CEMs and GPROs. Additionally, it describes curation rules and definitions to assign each identified chemical-protein interaction to any of the 10 classes, with detailed description listed below:

> 0. Part of: CEM that are structurally related to a GPRO: e.g. specific amino acid residues of a protein.
>
> 1. Regulator: CEM that clearly regulates a GPRO, but for which there is no further information on whether the regulation is direct or indirect.
>
> 2. Upregulator: CEM that increments a GPRO signal, without any insight on the mechanism.
>
> 3. Downregulator: CEM that decreases a GPRO signal, without any insight on the mechanism.
>
> 4. Agonist: CEM that binds to a receptor and alters the receptor state resulting in a biological response.
>
> 5. Antagonist: CEM that reduces the action of another CEM, generally an agonist. Many antagonists act at the same receptor macromolecule as the agonist.
>
> 6. Modulator: CEM that acts as allosteric modulator, compound that increases or decreases the action of an (primary or orthosteric) agonist or antagonist by combining with a distinct (allosteric or allotropic) site on the receptor macromolecule.
>
> 7. Cofactor: CEM that is required for a protein's biological activity to happen.
>
> 8. Substrate/Product: CEM that is both, substrate and product of enzymatic reaction.
>
> 9. NOT: This class should be used to define the NEGATIVE occurrence of a chemical-protein interaction, without providing any further information on the specific negative CHEMPROT class or class.

Please identity the CHEMPROT interaction to the best of your knowledge. Your answer should be classified into the following categories: [Part of, Regulator, Upregulator, Downregulator, Agonist, Antagonist, Modulator, Cofactor, Substrate/Product, NOT]. "

- SemEval: "You are an intelligent assistant to help recognize semantic relations between pairs of nomimals. For example, tea and ginseng are in an ENTITY-ORIGIN relation in "The cup contained tea from dried ginseng.". You will be given a piece of text, and Entity 1 and Entity 2 in the text for you to classify their semantic relation. The semantic relations are in the format of "entity1-entity2". The complete semantic relation inventory is given below:

> 0. Cause-Effect: An event or object (entity 1) leads to an effect (entity 2). Example: those cancers (entity 2) were caused by radiation exposures (entity 1)
>
> 1. Component-Whole: An object (entity 1) is a component of a larger whole (entity 2). Example: my apartment (entity 2) has a large kitchen (entity 1)
>
> 2. Content-Container: An object (entity 1) is physically stored in a delineated area of space (entity 2). Example: a bottle (entity 2) full of honey (entity 1) was weighed
>
> 3. Entity-Destination: An entity (entity 1) is moving towards a destination (entity 2). Example: the boy (entity 1) went to bed (entity 2)
>
> 4. Entity-Origin: An entity (entity 1) is coming or is derived from an origin (entity 2) (e.g., position or material). Example: letters (entity 1) from foreign countries (entity 2)
>
> 5. Instrument-Agency: An agent (entity 2) uses an instrument (entity 1). Example: phone (entity 1) operator (entity 2)
>
> 6. Member-Collection: A member (entity 1) forms a nonfunctional part of a collection (entity 2). Example: there are many trees (entity 1) in the forest (entity 2)

7. Message-Topic: A message (entity 1), written or spoken, is about a topic (entity 2). Example: the lecture (entity 1) was about semantics (entity 2)

8. Product-Producer: A producer (entity 2) causes a product (entity 1) to exist. Example: a factory (entity 2) manufactures suits (entity 1)

Please determine the semantic relation between entity 1 and entity 2 in the given text to the best of your knowledge. Your answer should be classified into the following categories: [Cause-Effect, Component-Whole, Content-Container, Entity-Destination, Entity-Origin, Instrument-Agency, Member-Collection, Message-Topic, Product-Producer]. "

- SMS: "You are an intelligent assistant to determine if a text message is spam or not spam (ham). Your answer should be classified into the following categories: [ham, spam]. "

**Response regularization prompt**   "You may think step by step, articulate point by point, or make conclusion from multiple evidences, but please always state the most likely label as your answer at the very begining of your response. You are encouraged to reflect on your response, but please keep in mind that a clear answer is always desired. Try to give a clear answer at your best guess even when you are not very sure, in which case any of your conserns or explanations should go after the most likely answer to the best of your knowledge. If you are very unsure about the answer and are not willing to explicitly state any label, please say 'unsure' at the very begining of your response. "

**Task instance prompt**

- Classification (for SMS):

  "Please classify the following example into the most likely category: [TEXT] "

- Relation extraction (for CDR, ChemProt, SemEval):

  "Please classify the following example into the most likely category: [TEXT] Entity 1 [ENTITY 1] Entity 2: [ENTITY 2] "

The complete prompt for querying the LLM is

**Problem setting prompt** + **Response regularization prompt** + **Task instance prompt**

### D.2   DYNAMIC PROMPTING

In dynamic prompting, we query another follow-up prompt after the LLM gives the initial out-of-the-box response. As this is an extension to our main experiments, we only implemented for the CDR relation extraction task. The follow-up prompts for the two dynamic prompting strategies are:

**Dynamic self-examination**   "Are you sure about your previous answer? If not, please give a new answer. Otherwise, please restate your previous answer. "

**Dynamic self-supervision**   "It is possible that the answer could be something else. Here are some evidences to help you figure out the right answer.

$$EvidencesFromSupervisionFunctions(x, \vec{\lambda}(x))$$

Are you sure about your previous answer? If not, please give a new answer. Otherwise, please restate your previous answer. "

$EvidencesFromSupervisionFunctions(x, \vec{\lambda}(x))$ contains evidences from all the supervision functions $\lambda_j(x) \neq 0$ that are triggered by the input instance $x$. Examples of evidence from the supervision functions are shown below. Note that each evidence will be provided only when the corresponding supervision function is triggered.

- "According to the Comparative Toxicogenomics Database, the relation between the given chemical-condition pair is listed, confirming the answer. "

- "According to the Comparative Toxicogenomics Database, the given chemical-condition pair "[ENTITY 1]-[ENTITY 2]" is listed that the chemical actually treats the condition, so the answer that [ENTITY 1] does not induce [ENTITY 2] is confirmed. "

- "According to the Comparative Toxicogenomics Database, the given chemical-condition pair "[ENTITY 1]-[ENTITY 2]" is listed that the chemical is typically present with the condition, which may confirm the answer if [ENTITY 1] induces [ENTITY 2]. "

- "Based on the expression [INDUCE PATTERN], it is likely that [ENTITY 1] induces [ENTITY 2]. "

- "Based on the expression [NOT INDUCE PATTERN], it is not likely that [ENTITY 1] induces [ENTITY 2]. "

- "Based on the expression [C TREATS D PATTERN], [ENTITY 1] actually treats [ENTITY 2]. , so it is not likely that [ENTITY 1] induces [ENTITY 2]. "

- "Based on the expression [CLOSE MENTION PATTERN], [ENTITY 1] is closely mentioned with [ENTITY 2], so they should be closely related. "

- "Based on the expression [DISEASE IMPROVE PATTERN], the disease [ENTITY 2] is actually improved, so it is not likely that [ENTITY 1] induces [ENTITY 2]. "

- "Based on the expression [INITIAL CONDITION PATTERN], [ENTITY 2] is the initial condition of the patient(s), so it is not likely that [ENTITY 1] induces [ENTITY 2]. "

- "Based on the expression [UNCERTAIN PATTERN], it is uncertain that [ENTITY 1] induces [ENTITY 2]. "

- "Based on the expression [INDUCED BY OTHER PATTERN], [ENTITY 2] is induced by other factors, so it is not likely that [ENTITY 1] induces [ENTITY 2]. "

- "[ENTITY 1] and [ENTITY 2] are not closely mentioned in the text, so it is not likely that [ENTITY 1] induces [ENTITY 2]. "

- "According to phrases like [WEAK EXPRESSION], there is no strong signal that [ENTITY 1] induces [ENTITY 2]. "

- "According to the text, another chemical is mentioned closer to [ENTITY 2] than [ENTITY 1], so it is not likely that [ENTITY 1] induces [ENTITY 2]. "

- "According to the text, another disease is mentioned closer to [ENTITY 1] than [ENTITY 2], so it is not likely that [ENTITY 1] induces [ENTITY 2]. "

