# OpenReview forum: "Automatic Calibration and Error Correction for Generative Large Language Models via Pareto Optimal Self-Supervision"
_ICLR.cc/2024/Conference — ICLR 2024 Conference Withdrawn Submission_

### Official Review · Reviewer_Mc5b · 2023-10-31

**Soundness:** 3 good
**Presentation:** 3 good
**Contribution:** 2 fair
**Rating:** 6
**Confidence:** 3

**Summary:**

This paper found that even weak supervision sources can correct the erroneous responses from large language models. Based on this finding, the paper proposes to correct the erroneous responses from large language models by first calculating a Pareto optimal learning assessed risk (POLAR) score to indicate error rate. Then according to each term in the weak supervision, the model will decide if more evidence needed to be add to the current prompt through the self-examination process. Finally, new responses will be generated given the enhance prompt. The POLAR score best aligns with the actual error rate in most cases on four datasets.

**Strengths:**

The paper proposes a systematic way to correct erroneous responses by first training a scoring function with weak supervision and then automatically enhancing the prompt based on the learned scoring function.

**Weaknesses:**

My biggest concern about this paper is I think it's over-claimed. This paper frames the learning objective to be a Pareto loss scalarizer, and also requires the Pareto scalarizer to be a convex function, which is too strong. In this case, the Pareto optimal is only a single point in the whole function space, which seems not satisfying the realistic scenario. A more natural way should be giving a different weight to different $l$, and then finding the Pareto optimal under specific weight. Thus, we have the Pareto frontier. According to different scenario, we may choose different Pareto optimal on the frontier. If in this paper's setting, then it seems the pipeline is just like first it learns some score functions using additional supervision signals. Then it uses these scoring function to enhance prompt, and finally it gets better responses based on better prompts, which seems no novelty to me.

**Questions:**

1. In Theorem 1, if G is always a convex function, then Theorem 1 always holds. I don't see any importance from Theorem 1. Does Theorem 1 still hold when G is a non-convex function?

2. In Equation 7, for generation task, there might be multiple correct  and incorrect responses such as the descriptions following the Negative in Figure 1 might be different. Taking one as the ground truth and all others as error ones, the calculated thePOLAR score might have some bias. Besides, how to calculate this score as the search space is huge right? I saw the author assumed that Y space is finite, but how to guarantee that as the description is different, leading to different LLM function score?

3. In Table 1 and Figure 1, what scalarizer is used?

---

> ### Author Response · Authors · 2023-11-20
>
> Dear Reviewer,
>
> Thanks for your comments and questions. We are making a revision to the paper for better clarity, and we appreciate the opportunity to clarify and elaborate on key aspects of the paper here.
>
> $\textbf{Regarding Pareto optimal Solution:}$
> - Based on our claim in Theorem 1, the harmonizer model $h^*$ identified via our approach approximates one of the possibly many Pareto optimal solutions. Indeed, different scenarios may favor different solutions on the Pareto frontier. We achieve this diversity through the selection of different Pareto scalarizers $G$, as outlined in our paper. Specifically, we explored three distinct Pareto scalarizers and the non-Pareto Chebyshev scalarizer for comparative analysis. Our empirical results, as shown in Table 2, suggest that the quadratic scalarizer is particularly effective when paired with the BERT harmonizer model. This scalarizer emphasizes challenging examples where the LLM and weak sources exhibit significant discrepancies, as briefly discussed following Definition 2.
>
> - We understand your concern regarding the need for reweighting among the LLM and weak sources. To address this, we provided two alternative strategies in Appendix B, designed to automatically adjust weights based on different scenarios, such as high dependency and significant differences in source accuracy. We hope these provide a good variety of Pareto optimality for different scenarios, addressing your concern.
>
> $\textbf{On the novelty of using scoring function:}$
> - We respectfully express our disagreement on "Finding scoring function from external weak sources and use them for calibration and correction has no novelty". Any calibration method, including early ones like Platt scaling, involves some form of scoring function. The challenge lies in deriving a robust scoring function with limited resources. Our approach uniquely tackles this by using weak supervision sources in the absence of ground truth labels, ensuring independence from the LLM and mitigating bias. We believe this is a reasonable approach to provide information that is independent from the LLM itself, which is essential in as a guardrail.
>
> - Our work's novelty is threefold:
>  1. We are pioneers in using weak supervision sources for LLM calibration and correction.
>  2. To our knowledge, this is arguably the first attempt to estimate confidence levels using only weak sources, a concept previously unexplored. We are excited to report our results that may inspire future work.
>  3. Our methodology for aggregating weak sources is innovative and distinct from prior works focused on label weighting.
>
>
> $\textbf{Responses to Specific Questions:}$
> - Question 1. Theorem 1 tells us the criteria for the function $G$ that are beyond mere convexity; it must also be strictly increasing element-wise (as per Definition 2). An illustrative counterexample is the Chebyshev/max scalarizer. Though convex, it fails to meet the strictly increasing condition, hence does not provide the guarantees outlined in Theorem 1. Our experiments support this, showing its relative underperformance compared to other Pareto scalarizers.
>
> - Question 2: Identifying hallucination and error in LLM is a broad and ongoing study area. In this study, we aim at solving the cases where the LLM response is mapped to a finite space, such as the categorical setting. While this doesn't encompass all LLM applications, it covers a substantial range including many important scenarios. This subset includes but but is limited to: GPT-4 evaluation on various exams, short answer QA evaluations, applications using LLM for structured outputs... For the example in Figure 1, one LLM response only maps to one answer.
>
> -  Question 3: We utilized the quadratic scalarizer in our experiments, as detailed in the 'Harmonizer Training' section of Section 4.
>
> We hope this addresses your concerns and provides further clarity on our paper. We are eager to answer any additional questions and hope you will consider revising your score with these clarifications.
>
> Best regards,
> Authors

---

> > ### Comment · Reviewer_Mc5b · 2023-11-22
> > **Reply to the rebuttal**
> >
> > Thanks for the response. I think my biggest concern has been addressed, so I raised my score from 5 to 6.

---

> > > ### Author Response · Authors · 2023-11-22
> > >
> > > Dear Reviewer,
> > >
> > > Thanks you so much for your feedback and decision! We appreciate you comments that help us to improve our paper in the revision.
> > >
> > > Best regards,
> > > Authors

---

### Official Review · Reviewer_Crzn · 2023-11-01

**Soundness:** 3 good
**Presentation:** 2 fair
**Contribution:** 4 excellent
**Rating:** 6
**Confidence:** 3

**Summary:**

# Summary
AUTOMATIC CALIBRATION AND ERROR CORRECTION FOR GENERATIVE LARGE LANGUAGE MODELS VIA PARETO OPTIMAL SELF-SUPERVISION

## What is the problem?
LLMs are very powerful and impactful, but suffer from significant rates of miscalibration resulting in possible hallucinations, which can be seen as erroneous responses made by the model with high confidence. These significantly degrade model performance in practice. This paper is concerned with designing a weak-supervision signal to produce calibrated estimates of the model's likelihood of error, and using those signals to improve model performance in the first place.

## Why is it impactful?
Reducing rates of hallucinations would have major impact on the utility and safety of LLMs.

## Why is it technically challenging/interesting (e.g., why do naive approaches not work)?
Model calibration is always challenging, let alone in the domain of LLMs in which we often work with complex, large input domains of free-text and arbitrary, under-defined labeling functions.

## Why have existing approaches failed?
Existing approaches have not (to the best of my knowledge) as rigorously examined leveraging weak supervision as is explored in this work.

## What is this paper's contribution?
This paper uses pareto-optimization techniques to maximally harmonize a set of weak-supervision signals and an LLMs output to produce a score estimating the model's uncertainty in its predictions.

## How do these methods compare to prior works?
There are some existing works that should be referenced in this paper that seem to be missing:
  1. https://arxiv.org/abs/2104.08455
  2. https://ojs.aaai.org/index.php/AAAI/article/view/26596
  3. https://dl.acm.org/doi/abs/10.1145/3583780.3614905?casa_token=QsAQ_cJ35qwAAAAA:m4oawA8tORNEv3Nkn3zON32LJtqmBBkQgoL7dOGC5IeeNCt59-tcyoHUDsTcs5O0lzerxmtXSSJi

This is a very fast moving space, so there are some more recent works that should also be cited in the final draft (though they weren't out when this paper was submitted):
  1. https://arxiv.org/abs/2309.11495
  2. https://arxiv.org/abs/2308.11764
  3. https://arxiv.org/abs/2310.03951
  4. https://arxiv.org/abs/2310.00259
  5. https://arxiv.org/abs/2309.02654

## How do they validate their contributions?
They show that these uncertainty estimates are well calibrated and that using them via a self-supervision technique can significantly improve model performance.

**Strengths:**

## Key Strengths (reasons I would advocate this paper be accepted)
  1. This is a critical problem area to study.
  2. This approach is well thought out and the empirical results are compelling.
  3. This approach seems extensible to other sources of weak supervision as well.

**Weaknesses:**

## Key Weaknesses (reasons I would advocate this paper be rejected)
  1. This paper has major issues with clarity. You have a lot of typos / odd grammatical issues and sentence structures that make it hard to understand. E.g., "Unlike task-specific models, there lack an effective method..." in the abstract. Even beyond these, there is some major missing information throughout in important sections. For example, on the top of page 4 (from the end of Proposition 1 to the start of Proposition 2), you have a paragraph that contains a lot of statements illustrating potential problems and arguing that certain things are therefore needed with little to no explanation of these key ideas and why we should believe they are true. E.g., why does Proposition 1 show a path towards calibrating an LLM via distillation? How does this show the need of external signals? etc. In addition, some mathematical statements are not sufficiently explicitly stated to be correct as written, which need to be corrected for sure.
  2. Your proposition 2 is not sufficiently well specified to be true. You need to expand on what "weak independent signal ensemble" means to eliminate the case that the weak ensemble contains no new information relative to your LLM itself (and therefore cannot contribute meaningful improvements over the LLM). Also your notation for the function $\psi$ is inconsistent, as $\psi$ is listed as a function of both the functions $\Lambda$ and $w$ as well as of their outputs. And you need to clarify why your $w$ is a continuous estimator, not a categorical, given you work in the categorical space in other settings.
  3. The inference you make in the second sentence of 3.3 is invalid given that you don't know that these "weakest supervision sources such as reg-ex or knowledge base" are actually independent of the LLM. I know you go back and state this in the next sentence or two, but if you're going to acknowledge it there, just be specific and clear from the outset.
  4. Isn't your "pareto optimization scalarizer" functionally reducing all your weak supervision signals to a single signal, thereby suffering the same concerns that you state the existing works will suffer by combining loss terms via a weighted sum?

**Questions:**

## Minor Questions
  1. Do you iterate repeatedly in Algorithm 2 until your POLAR score is below $\delta$? Or do you just do the process once.

## Things I need to see to improve my score:
While I listed only a small amount of content in the strengths section and a lot in the weaknesses section, my overall recommendation here is a marginal accept. I feel that this contribution, and the results reported, suggest that this contribution is possibly of sufficient significance to warrant acceptance despite its somewhat severe issues with clarity and presentation.

What I vigorously want to see in the revision to further improve my score is a significant revision to improve the quality of the writing (eliminate typos, grammatical errors, etc.), clarify all of the sections of the text, especially w.r.t. the methodology and details behind the experiments, and formal, correct, full statements behind the mathematical claims made in the work.

---

> ### Author Response · Authors · 2023-11-22
>
> Dear Reviewer,
>
> Thank you for your thorough review and constructive feedback on our manuscript. We sincerely appreciate your acknowledgement of our contribution. The suggestions you made to improve our paper presentation are highly valuable, and we have made significant revisions to address your concerns, particularly regarding clarity and the specificity of our theoretical claims. Here is a summary of the key revisions and responses to your questions:
>
> 1. $\textbf{Clarity and Writing Quality:}$ We have carefully revised the manuscript to eliminate typos and grammatical issues. We have also restructured sentences for better clarity and coherence, particularly in sections where the methodology and experimental details are explained. This includes clarifying the role and the limitations of Propositions 1 and 2, and providing a more explicit explanation of our approach in the end of Section 3.3. Although there are many changes in the paper, for your time and convenience we want to highlight one example of modification for your reference. This is the part after Proposition 1 (distillation from imperfect source promotes calibration), and before Proposition 2 (noisy weak signals helps accurate LLM):
>
> "(After Proposition 1) ... Muller et al. (2019) empirically demonstrated
> that label smoothing improves or at least doesn’t harm model calibration compared to standard
> supervised learning. Therefore, Proposition 1 suggests that access to ground truth labels is not
> necessary for obtaining well-calibrated outputs. If LLM errors were uniformly “random”, we could
> directly use it as a teacher model to distill a small network on the specific task for calibration.
>
> However, LLM errors are rarely “random”. Deterministic functions typically err on specific types of
> inputs. Distilling a model solely from the LLM could lead to overfitting, thus unable to signal errors
> in the teacher LLM. Our solution is to integrate other weak supervision functions as external signals
> independent of the LLM. As each supervision source makes different types of errors on different
> examples, their ensemble approximates a “random” scenario... (Before Proposition 2)"
>
> 2. $\textbf{Theoretical Clarification:}$ Regarding Proposition 2, we have expanded the explanation of "weak independent signal ensemble" to clarify that these signals provide information independent of the LLM. We have also revised the notation for the function "$\psi(\Lambda, w)$" to "$\psi(\Lambda(x), w(x))$" to avoid confusion. Additionally, we now explicitly state why the weak supervision signal ensemble $\omega(x)$ is modeled with a continuous random variable in Proposition 2.
>
> 3. $\textbf{Section 3.3 Clarification}$ We now explicitly states in this section that our main theoretical results takes a leap from the inspirational propositions. We also added a brief summary for the theoretical part at the end of Section 3.3 to discuss the connections between the theoretical results and their limitations. Here is the content for your reference:
>
> "Let’s make a brief theoretical summary here. If the LLM makes uniformly “random” errors, simply distilling a smaller network would give a calibration model. Since this is not usually the case, we incorporate multiple independent supervision functions alongside the LLM, fitting a harmonizer model to all simultaneously to prevent overfitting to the LLM. Theoretically, we approximate a harmonizer model that is Pareto optimal. However, due to the complexity of modeling the errors made by the LLM and supervision functions, we cannot offer further theoretical guarantees at this point. We empirically demonstrate the calibration capability of the harmonizer model in Section 4."
>
> 4. $\textbf{Pareto Optimization Scalarizer:}$ We have further clarified the concept of the Pareto optimization scalarizer in our framework. We clarified that while it does aggregate multiple supervision signals into a single harmonized output, this process is fundamentally different from a simple weighted sum because of Jensen's inequality. One example is the quadratic scalarizer, which automatically emphasizes on the harder examples with larger overall loss. The trade-offs between different supervision sources is handled in the Pareto optimal manner, thereby avoiding the pitfalls of traditional weighted sum approaches.
>
> (To be continued...)

---

> ### Author Response · Authors · 2023-11-22
>
> (... Continued from the previous comment...)
>
> 5. $\textbf{Dynamic Prompting Iteration in Algorithm 2:}$ The POLAR-assisted dynamic prompting process in Algorithm 2 is typically executed once per instance. If we take the threshold $\delta = 0.5$, after one iteration of correction it must be below $\delta$ already. However, one can re-fit the harmonizer using the updated LLM response and do it again. In our experiments, we didn't observe additional significant improvement on performance, likely due to the information limit.
>
> 6. $\textbf{Additional References and Comparisons:}$ We have added the suggested recent references to our manuscript and discussed how our work relates to these developments in the field. Specifically, we have also included a comparison with another weak supervision model in our experimental evaluations (Table 1) to demonstrate the advantages of our approach.
>
> 7. $\textbf{Experimental Details:}$ The details behind the experiments have been expanded for greater clarity. This includes more information on the datasets used, the specifics of the implementation of the harmonizer, and the rationale behind the choice of our evaluation metrics. Explanations of the benchmark comparison methods are also expanded.
>
> Thank you again for your time and valuable comments, which did put our work and paper into a better state! We believe these revisions have significantly strengthened the manuscript, addressing the key concerns you raised. Hope that the improvements in clarity, presentation, citation, and extra experiments adequately address your feedback and make the paper a strong candidate for acceptance.
>
> Thank you once again for your valuable feedback, and we look forward to your further thoughts!
>
> Best regards,
> Authors

---

> > ### Comment · Reviewer_Crzn · 2023-11-22
> > **Thank you for your response!**
> >
> > Thank you for your detailed response! Unfortunately, as the response only came in this morning and the discussion period closes tonight, I have not had the ability to go through your new revisions to sufficient degree to see if my score should increase further, but thank you regardless for these comments!
> >
> > P.S. I'd also recommend in the future indicating new or changed text in your updated PDF via a different color or some such, to make it easier on reviewers to see changes.

---

> > > ### Author Response · Authors · 2023-11-22
> > >
> > > Dear Reviewer,
> > >
> > > Thanks for your response. As we were doing a major revision in terms of writing, and implemented new experiments per other reviewer's request, it took relatively longer time for us to present the updated paper. We want to thank you again for your valuable input that helped us to deliver a better version.
> > >
> > > Correct me if I am wrong, I think the critical deadline tonight is for us to make any revision and responses, and we can see your new comments afterwards. We are grateful for your time and effort in your initial inputs already, but please feel free to make any further comments!
> > >
> > > P.S. We highlighted some of the major changes in our initial response to relief your effort going back into the paper. We considered marking the changes in the PDF, but since ICLR author guide says there is the pdfdiff feature on the reviewer side, we decided to keep the revision in the normal format. We highly value your time and do want to make it easier for you.
> > >
> > > Thanks you again!
> > >
> > > Best regards,
> > > Authors

---

> > > ### Author Response · Authors · 2023-11-23
> > >
> > > Dear Reviewer,
> > >
> > > We just uploaded a new revision PDF with all significant changes highlighted in red. We want to express our gratitude again for your time and effort, and your valuable input to our paper. We understand there is limited time in the discussion period, just want to present our revised work to you in a better state as your comments significant helped us improve it. Thanks!
> > >
> > > Best regards,
> > > Authors

---

### Official Review · Reviewer_trZj · 2023-11-01

**Soundness:** 2 fair
**Presentation:** 3 good
**Contribution:** 2 fair
**Rating:** 6
**Confidence:** 4

**Summary:**

It is important to detect when Large Language Models (LLMs) hallucinate. Namely, we'd like to know the level of confidence of an LLM, e.g. the quantity $\Pr(\Lambda(x) = y | x)$. To better calibrate LLMs, this paper leverages external sources of supervision to estimate this confidence score. The paper frames this as a multi-objective problem, training a harmonizer that minimizes discrepancies between the LLM's output and the supervision functions. One can then derive the POLAR score from the harmonizer, which serves as an estimate of the actual risk associated with each data sample. The paper also presents a way to use the POLAR score for dynamic prompting, where a prompt that is likely overconfident is patched with feedback/reflection from the supervision functions. Empirically, the paper demonstrates a strong correlation between the POLAR score and the true error rate obtained from gold labels. It also shows the effectiveness of using the POLAR score to rectify errors in LLMs, compared to approaches based on standard weak supervision (Snorkel) or training a smaller model on the LLM output (Distillation).

**Strengths:**

Originality:
- Pareto-optimal multi-objective approach is novel compared to other weak supervision techniques.

Quality:
- Strong empirical validation
- Dynamic prompting appears to work very well

Clarity:
- Paper's results and problem setup were easy to follow.

Significance:
- The paper touches on an important issue in using LLMs and gets around approaches that can include the model's own biases. Beyond calibrating, the dynamic prompting approach shows a workflow in which error detection and model editing can be carried out effectively.

**Weaknesses:**

Originality:
- Weighting dilemma is not something new in weak supervision. A major goal is rather than using one weight per supervision function across the entire dataset, we'd like to have input-specific weight parameters (such as https://arxiv.org/abs/2107.02233).

Quality:
- Limitations of theoretical results are not discussed well. For instance in proposition 2, the existence of a function doesn't imply that this function class is how $\Lambda(x), w(x)$ is aggregated.
- The paper could be made stronger by comparing to weak supervision methods that also solve the weighting dilemma, like the paper linked above I think.

**Questions:**

- Can you discuss limitations of theoretical results?
- See my question about comparing to additional weak supervision methods.

---

> ### Author Response · Authors · 2023-11-22
>
> Dear reviewer,
>
> We sincerely appreciate your insightful comments and feedbacks. We have undertaken a major revision of our paper, taking into account your valuable suggestions. Below are the key areas of improvement:
>
> 1. $\textbf{Enhanced Theoretical Section:}$
> - We have revised the theoretical section for enhanced clarity, adding detailed explanations and discussions about the implications and limitations of our propositions and theorems. We acknowledge that our propositions are more inspirational in nature and may contain unrealistic assumptions (e.g., non-random LLM errors, independence of weak supervision sources, and an insufficient number of weak sources for CLT approximation). For main results, we provide theoretical guarantee with Theorem 1 free from these assumptions, and we demonstrate the empirical effectiveness of our approach.
>
> - A new paragraph at the end of Section 3.3 now provides a summary and discusses the connection between the theoretical results and their limitations. Here is the added content for you reference:
>
> "Let’s make a brief theoretical summary here. If the LLM makes uniformly “random” errors, simply distilling a smaller network would give a calibration model. Since this is not usually the case, we incorporate multiple independent supervision functions alongside the LLM, fitting a harmonizer model to all simultaneously to prevent overfitting to the LLM. Theoretically, we approximate a harmonizer model that is Pareto optimal. However, due to the complexity of modeling the errors made by the LLM and supervision functions, we cannot offer further theoretical guarantees at this point. We empirically demonstrate the calibration capability of the harmonizer model in Section 4."
>
> 2. $\textbf{Benchmark Experiments with WeaSEL:}$
> - In response to your suggestion, we have included the End-to-end weak supervision paper in our citations. This paper presents an nice framework in weak supervision, assigning instance-specific weights in an end-to-end manner.
>
> - We conducted benchmark experiments using the WeaSEL model and have included these results in our comparison Table 1. While WeaSEL shows commendable performance on specific tasks and LLMs (e.g., calibrating GPT-3.5 on the CDR relation extraction), it does not consistently deliver across all scenarios. Notably, it performs optimally with GPT-3.5 and in situations with a limited number of supervision functions. Compared to the Snorkel model, WeaSEL exhibits superior performance in most cases, likely due to its greater expressiveness and instance-specific weighting. For your convenience, we provide the extended comparison table in the next comment.
>
>
> We want to thank you again for bringing the comments and suggestions. We believe these revisions substantially strengthen our paper, providing a more nuanced theoretical discussion and adding robust empirical comparisons. We hope this addresses your concerns and would be grateful for any further comments or suggestions. Your consideration in revising the rating is greatly appreciated.
>
> Best regards,
> Authors

---

> ### Author Response · Authors · 2023-11-22
> **Extended Experimental Results with Extra Benchmark Comparison**
>
> |               | POLAR             |            | Snorkel          |            | WeaSEL           |            | Majority vote    |            | LLM Distilled    |            |
> |---------------|-------------------|------------|------------------|------------|------------------|------------|------------------|------------|------------------|------------|
> |               | ECE               | \(R^2\)    | ECE              | \(R^2\)    | ECE              | \(R^2\)    | ECE              | \(R^2\)    | ECE              | \(R^2\)    |
> | **CDR**       |                   |            |                  |            |                  |            |                  |            |                  |            |
> | GPT-4         | **0.043**         | **0.890**  | 0.167            | 0.299      | 0.146            | 0.387      | 0.145            | 0.348      | 0.164            | 0.592      |
> | GPT-3.5-turbo | **0.046**         | 0.934      | 0.164            | 0.320      | 0.081            | **0.942**  | 0.182            | 0.540      | 0.222            | 0.037      |
> | Text-davinci-3| **0.055**         | **0.907**  | 0.154            | 0.371      | 0.135            | 0.877      | 0.149            | 0.450      | 0.089            | 0.896      |
> | **ChemProt**  |                   |            |                  |            |                  |            |                  |            |                  |            |
> | GPT-4         | **0.035**         | **0.934**  | 0.182            | 0.510      | 0.278            | 0.885      | 0.233            | 0.244      | 0.216            | 0.766      |
> | GPT-3.5-turbo | **0.048**         | **0.944**  | 0.228            | 0.625      | 0.219            | 0.922      | 0.282            | 0.031      | 0.159            | 0.845      |
> | Text-davinci-3| **0.051**         | **0.917**  | 0.218            | 0.700      | 0.213            | 0.846      | 0.279            | 0.307      | 0.196            | 0.825      |
> | **SemEval**   |                   |            |                  |            |                  |            |                  |            |                  |            |
> | GPT-4         | 0.079             | 0.916      | 0.068            | 0.714      | 0.612            | 0.626      | 0.115            | 0.379      | **0.063**        | **0.947**  |
> | GPT-3.5-turbo | **0.047**         | **0.963**  | 0.150            | 0.821      | 0.345            | 0.890      | 0.277            | 0.208      | 0.108            | 0.757      |
> | Text-davinci-3| 0.067             | **0.950**  | 0.119            | 0.796      | 0.455            | 0.784      | 0.242            | 0.396      | **0.065**        | 0.936      |
> | **SMS**       |                   |            |                  |            |                  |            |                  |            |                  |            |
> | GPT-4         | 0.014             | **0.980**  | 0.244            | 0.089      | 0.409            | 0.345      | 0.588            | 0.091      | **0.013**        | 0.977      |
> | GPT-3.5-turbo | **0.041**         | **0.963**  | 0.075            | 0.202      | 0.286            | 0.731      | 0.148            | 0.006      | 0.131            | 0.775      |
> | Text-davinci-3| **0.023**         | 0.943      | 0.201            | 0.053      | 0.420            | 0.238      | 0.325            | 0.091      | 0.033            | **0.956**  |
>
> As shown in the table, for all three LLMs across all four tasks, the proposed POLAR score consistently outperforms other methods. Among the baseline methods, Snorkel, WeaSEL, and LLM distilled model can achieve top or close-to-top performance in some cases under specific metric, but lack the consistency to deliver stable calibration for different LLMs on different tasks. In comparison, the proposed POLAR score is consistently well-calibrated to the true error rate.
>
> While WeaSEL shows commendable performance on specific tasks and LLMs (e.g., calibrating GPT-3.5 on the CDR relation extraction), it does not consistently deliver across all scenarios. Notably, it performs optimally with GPT-3.5 and in situations with a limited number of supervision functions. Compared to the Snorkel model, WeaSEL exhibits superior performance in most cases, likely due to its greater expressiveness and instance-specific weighting.
>
> We hope this extra benchmark comparison strengthen the effectiveness of our approach. We appreciate any of your comments and questions, and your consideration for elevating the score.
>
> Best regards,
> Authors

---

### Official Review · Reviewer_mUxK · 2023-11-06

**Soundness:** 3 good
**Presentation:** 3 good
**Contribution:** 2 fair
**Rating:** 5
**Confidence:** 2

**Summary:**

Although the development of large language models (LLMs) is in full swing, the phenomenon of hallucinations has been an issue that we cannot ignore, as it poses a significant challenge to the reliability of LLMs in real-world scenarios. Given the lack of an effective method to calibrate the confidence level of LLM responses to reveal potential errors, this paper proposes a novel framework for LLM calibration using Pareto optimal self-supervision. Specifically, it leverages available programmatic supervision to systematically calibrate LLM responses by producing a risk score for every LLM response without any additional manual efforts. Experimental results show the proposed risk score is consistently well-calibrated to the probability of LLM error. On this basis, a dynamic prompting strategy has been further developed to automatically correct LLM errors, boosting GPT-4 results past SOTA supervised results on challenging evaluation datasets.

**Strengths:**

- It's a meaningful attempt to calibrate the confidence level of LLM responses to indicate potential errors. Combining information from both the LLM response and other supervision sources, the paper presents a flexible framework that optimizes a harmonizer network $h(x)$ to assist in assessing the risk of LLMs' outputs using Pareto optimal learning.
- The article is rich in theoretical arguments, explaining how calibrating LLM and detecting errors can be achieved by learning a harmonizer network using the Pareto optimal theory.
- Experiment results demonstrate that the proposed Pareto optimal learning assessed risk (POLAR) score is effective in estimating LLM error probability, and the designed dynamic prompting strategy boosts a significant accuracy improvement for off-the-shelf LLMs without any manually labeled training data.

**Weaknesses:**

- The proposed framework is restricted to tasks where the desired output space $\mathcal{Y}$ is finite, which can be impractical for real-world applications when there is no definitive answer in a finite set, e.g., recommending websites for learning languages.
- I am sorry, but I don't think the two intuitions behind how calibration can be achieved (mentioned by the authors in the introduction) are easy to understand, perhaps due to my lack of expertise. They look like two conclusions that other works have confirmed. Are there any references?
- I do not see a natural connection between **Proposition 1**, **Proposition 2** and Pareto optimal learning. According to the **Proposition 1**, fitting a model $h(x)$ to $Λ(x)$ is equivalent to training on ground truth labels with label smoothing. However, I do not find that the author mentioned using the "label smoothing" technique while optimizing the harmonizer network $h(x)$. Also, I'm not sure how this is solved by using Pareto optimal learning.

**Questions:**

- As a major part contributing to the multi-objective optimization, the supervision functions $\lambda_j$ do not seem to be clearly presented.
- In the **Definition 2** (Pareto scalarizer), what is the difference between $G(\mathcal{l}_0, \cdots, \mathcal{l}^{'}_j, \cdots, \mathcal{l}_m)$ and $G(\mathcal{l}_0, \cdots, \mathcal{l}_j, \cdots, \mathcal{l}_m)$, as far as I understand, they have the same $m$ objectives?
- So is the Pareto scalarizer actually equivalent to the uniform weighting scheme?
- How are the two metrics based on  POLAR score, i.e., expected calibration error (ECE) and $R^2$ calculated?
- For every baseline method, is the error rate calculated by comparing the estimated class probabilities with ground truth labels?
- From Figure 3(a), it seems that we can always use the dynamic self-supervision prompting strategy to achieve a decent error rate?

**Details Of Ethics Concerns:**

No ethical issues found.

---

> ### Author Response · Authors · 2023-11-22
>
> Dear Reviewer,
>
> Thank you for your insightful feedback and questions. We greatly appreciate your comments and have undertaken a comprehensive revision to address your concerns. Below is a summary of how we have addressed each point, and our response to the questions.
>
> **Response to Weaknesses**
>
> 1. **Applicability to Real-World Tasks with Non-Finite Output Spaces:** We agree with you and recognize the limitation of our framework being applicable primarily to tasks with finite output spaces. As identifying and correcting LLM error is a challenging problem and an emerging field, we chose to be focus on and tackle a smaller problem. What we hope from our paper is not only the introduction of the proposed method, but also theoretical and empirical results that can inspire future work. In that sense, targeting a smaller problem helps us dig deeper.
>
> 2. **Clarity on Theoretical Intuitions:** We have revised the introduction to provide clearer explanations of the intuitions behind our approach. The two intuitions are proved in Proposition 1 and 2 respectively. In short, if the LLM error is uniformly random, the distilled model from it serves as a good calibration model. Introducing multiple external supervision sources diversifies the output, avoiding the distilled model being completely bias by the LLM and not able to signal its error.
>
> 3. **Connection Between Propositions and Pareto Optimal Learning:**
>
> To clarify the relationship between Propositions 1 and 2 and Pareto optimal learning, we have added a detailed explanation in Section 3.2. We also added brief summary of the connections between the theoretical results, as well as their limitations. The content is below for your reference:
>
> "Let’s make a brief theoretical summary here. If the LLM makes uniformly “random” errors, simply distilling a smaller network would give a calibration model. Since this is not usually the case, we incorporate multiple independent supervision functions alongside the LLM, fitting a harmonizer model to all simultaneously to prevent overfitting to the LLM. Theoretically, we approximate a harmonizer model that is Pareto optimal. However, due to the complexity of modeling the errors made by the LLM and supervision functions, we cannot offer further theoretical guarantees at this point. We empirically demonstrate the calibration capability of the harmonizer model in Section 4."
>
> 4. **Label Smoothing:** In Proposition 1, label smoothing is implicitly achieved by fitting to the imperfect LLM output. This technique is not adopted for training, but as a theoretical concept to justify our approach. We have made the explanation for this clearer in the revision to avoid confusion.
>
> **Response to Questions:**
>
> 5. **Presentation of Supervision Functions:** We have revised the manuscript to provide a more detailed explanation of the supervision functions. In short, they are any programmatic supervision sources, such as keywords, regex, knowledge base, that outputs noisy heuristic labels on the input. We explained the choice of the supervision function in the experimental part, and also discussed their contribution to the overall calibration performance.
>
> 6. **Objectives in Pareto Scalarizer:** The difference between $G(\ell_0, \cdots, \ell_j', \cdots, \ell_m)$ and $G(\ell_0, \cdots, \ell_j, \cdots, \ell_m)$ is $\ell_j'$ and $\ell_j$. The requirement for $G$ is, for any $\ell_j' < \ell_j$, there must be $G(\ell_0, \cdots, \ell_j', \cdots, \ell_m) < G(\ell_0, \cdots, \ell_j, \cdots, \ell_m)$, i.e. strictly increasing (but not necessarily continuous).
>
> 7. **Clarification on Pareto Scalarizer and Weighting Schemes:** The Pareto scalarizer in our experiment is a nonlinear function aggregating the losses, thus different from a uniform weighting scheme. When performing SGD types of optimization, Jensen's inequality makes the nonlinearity take effect. For example, the quadratic scalarizer, which is the one we experimented to be most effective, automatically emphasize on the hard examples where the LLM and the other sources have larger disagreement. In our revision, we highlight its unique role in balancing multiple objectives in a harmonized manner.
>
> 8. **Computation of POLAR Score Metrics:**
> - ECE: Split the examples into 10 bins according to their POLAR score, e.g. 0-10%, 10-20%, etc. For the examples in each bin, calculate the real LLM error rate using the ground truth labels. Compare the difference between the POLAR score and the true LLM error rate, and take expectation over all the examples.
> - $R^2$: Ranks the LLM responses into POLAR score bins each with 100 examples, and plot the average POLAR score and true error rate for each bin. The R2 is calculated from the (POLAR - error rate) scatter plot.
>
> (To be continued...)

---

> ### Author Response · Authors · 2023-11-22
>
> (... Continued from last comment...)
>
> 9. **Error Rate Calculation in Baseline Methods:** Yes, the error rate is indeed calculated by comparing the estimated class probabilities from the baseline model with ground truth labels.
>
> 10. **Dynamic Self-Supervision Prompting Strategy:** Yes, as long as the supervision functions are independent from the LLM (thus provide extra information), we can always improve the error rate for the high POLAR score examples. Once the error got corrected, because the POLAR score already dropped below the threshold, re-prompting doesn't further improve the results. This is consistent with the information limit available from the supervision functions.
>
> We believe our revisions and responses address the key points of your feedback and strengthen the contribution of our paper. We hope that our responses and the improvements made will lead to a reconsideration of the rating.
>
> Thank you again for your valuable input! We look forward to any further suggestions or questions you may have.
>
> Best regards,
> Authors

---

> > ### Comment · Reviewer_mUxK · 2023-11-23
> > **Thanks for the response**
> >
> > I thank the authors for their detailed and thorough explanations! I think part of my concerns have been addressed, before I decide whether or not to change my rating, I need to check carefully the correspondings changes in the new revision, which I would recommend to highlight using a different color.

---

> > > ### Author Response · Authors · 2023-11-23
> > >
> > > Dear reviewer,
> > >
> > > Thanks for your response! We appreciate your time and effort in helping us improve our work. An updated revision with color highlighting is posted, with all significant changes highlighted in red. Hope it helps you in your review. We would appreciate it greatly if our modification based on your comments makes a higher rating of the paper!
> > >
> > > Best regards,
> > > Authors

---

### Meta-Review · Area_Chair_Padb · 2024-01-06

**Metareview:**

The authors propose a method for using a set of noisy 'weak' supervision signals to calibrate a confidence estimation for the output of a LLM system. Operationally, this work builds on results from data programming (e.g., Snorkel) based on the assumption that agreement between the LLM output and any weak supervision signal is better than random to propose a Pareto optimization framework that estimates a confidence of the LLM output using a harmonizer network -- without the need for manual 'gold' labels. The resulting score is referred to as the Pareto optimal learning assessed risk (POLAR) score. Empirical results on difficult relation extraction and classification tasks demonstrate that the POLAR score correlates with the LLM error rate and can be used for automatic error correction to derive strong empirical results for the target applications.

Consensus strengths identified by reviewers regarding this submission include:
- Estimating the output confidence of LLMs is a clear weakness that has attracted recent attention. The proposed approach draws from recent data programming work and makes sufficient LLM targeting modification to develop a well-principled and performant solution for the target tasks.
- By drawing from data programming solutions and Pareto optimization, this work inherits solid theoretical footing that can likely be expanded with additional examination of the harmonizer network and Pareto scalarizers.
- The empirical results successfully demonstrate that POLAR is effective in estimating LLM error probability and the corresponding dynamic prompting strategy for self-initiated error correction boosts a significant accuracy improvement for off-the-shelf LLMs without any manually labeled training data.
- Overall, the paper is clearly written and easy to understand.

Conversely, consensus limitations included:
- The primary concern seemed to be that the paper is limited to classification problems (or at least finite-class prediction problems). Thus, there is a mismatch between the claim of 'hallucination' and 'misclassification'. Connecting the findings in this paper to general hallucinations in generation tasks still requires a significant amount of work.
- There were many theoretical and conceptual clarification questions. These were largely addressed during the rebuttal stage and incorporated into the writing. However, some of these concerns remained during discussion.
- The novelty is primarily driven by findings from the 'multiple weak labeling' setting, best exemplified by the data programming paradigm. I agree with this concern as this paper isn't really specific to LLMs (between the harmonizer method and this being a classification problem). The only thing specific to LLMs is the assumption that we have one strong output (the LLM) and multiple 'weak' labeling that are correlated more than random and that this can be used to reprompt (which has an analogue in flipping labels). This was partially addressed with additional baselines, but there remains a conceptual concern regarding novelty and LLM-specific applicability (since this is the premise of the paper).

Overall, this is an important direction to study in the context of LLMs. However, by exclusively addressing classification problems, this work doesn't address the most common usages of LLMs (and generative AI in general). I believe this is a good start and would be of interest to the community, but more as inspiring additional work than being directly implemented as part of a general LLM framework.

**Justification For Why Not Higher Score:**

This work is more about data programming and confidence estimation from multiple weak annotators than LLM output calibration. This is most clearly reflected in the exclusive study of classification problems.

**Justification For Why Not Lower Score:**

N/A

---

### Decision · Program_Chairs · 2024-01-16

Reject